# Brief Communication: Rapid ~335 $10^6$ m$^3$ bed erosion after detachment of the Sedongpu Glacier (Tibet)

Andreas Kääb[1], Luc Girod[1]

[1]Department of Geosciences, University of Oslo, Oslo, 0316, Norway

*Correspondence to*: Andreas Kääb (kaeaeb@geo.uio.no)

**Abstract.** Following the 130±5 $10^6$ m$^3$ detachment of the Sedongpu Glacier, south-eastern Tibet, in October 2018 the Sedongpu valley, which drains into the Yarlung Tsangpo (Brahmaputra) River, underwent rapid large-volume landscape changes. Between December 2018 and 2022 and in particular during summer 2021, an enormous volume of in total ~335±5 $10^6$ m$^3$ was eroded from the former glacier bed, forming a new canyon of up to 300 m depth, 1 km width and almost 4 km length. The 2021 erosion peak happened through massive but still gradual retrogressive erosion into the former glacier bed. Several rock-ice avalanches of in total ~150±5 $10^6$ m$^3$ added to the total rock, sediment and ice volume of over 600 $10^6$ m$^3$ (0.6 km$^3$) that were exported from the basin since around 2017. The recent erosion volumes at Sedongpu are by order of magnitude equivalent to the average annual denudation volume of the entire Brahmaputra basin upstream of the location where the river leaves the Himalayas. This high-magnitude low-frequency event illustrates a potential for rapid post-glacial landscape evolution and associated hazards that has rarely been observed and considered at such high intensity so far.

## 1 Introduction

Current retreat of mountain glaciers uncovers large areas of unconsolidated sediments that were previously held under ice and thus protected against most direct impacts by weather and climate. After glacier disappearance these newly uncovered areas are subject to erosion processes at different time scales, from slow century-long background denudation to rapid mass loss by sediment-rich flows such as debris flows (Ballantyne, 2002; Carrivick and Heckmann, 2017; Williams and Koppes, 2019). Recent retreat of mountain glaciers is likely contributing to increased debris-flow activity and thus climate-change driven increase in mountain hazards (Zimmermann and Haeberli, 1992; Ballantyne, 2002; Hock et al., 2019). Glaciers retreat comparably slowly, over many years to decades, so that the full potential erosion volume of formerly subglacial soft-beds is difficult to estimate from the usual successive bed erosion that follows gradual glacier retreat. The reconstruction of original subglacial sediment volumes or eroded volumes is difficult at the locations of former valley glaciers, unless significant parts of the glacier bed are still preserved.

Rare but massive glacier detachments offer a unique natural experiment to investigate what happens to a glacier bed once the glacier above it is rapidly removed, giving an indication on the maximum erosion potential that might else be mobilized over

longer time scales of gradual glacier retreat. During such large-volume detachments, entire low-angle valley glaciers are removed within minutes (Kääb et al., 2018; Jacquemart et al., 2020; Kääb et al., 2021), leaving their beds suddenly exposed to weather and climate impacts. Low-angle glacier detachments seem to be associated with particularly soft, and thus easily erodible basal sediments (Gilbert et al., 2018; Kääb et al., 2021). A very recent and at the same time one of the largest glacier detachments known to date is the 2018 Sedongpu, Tibet, event. In this study, we investigate the development of the glacier

bed after glacier removal in order to draw conclusions about bed stability and erosion potential, the landscape evolution in response to glacier loss, and the associated hazard potential. We summarize key site information on the 2018 glacier detachment, and quantify the glacier-bed volume changes and other landscape changes in the basin until November 2022.

## 2 Study site

At the time before its detachment, the tongue of the Sedongpu Glacier was situated in south-eastern Tibet (29.80° N, 94.92°

E), at an elevation of about 3700 m asl. (former elevation range of entire glacier ~3550 – ~7200 m asl., including contributing hanging glaciers and snow/ice fields), just 4 km north of the Yarlung Tsangpo River (Tibetan name for the upper reaches of Brahmaputra), which the Sedongpu valley joins at an elevation of around 2700 m asl. The highest point of the Sedongpu catchment is the Gyala Peri peak (7294 m asl.; Fig. 1). The study site, in particular the western flank of Gyala Peri represents extreme topography in terms of relief and overall slope angles, for instance between 40° and 45° measured

from Gyala Peri down to the detached glacier tongue, or towards south to the Yarlung Tsangpo. The Gyala Peri and neighbouring Namcha-Barwa massifs have exceptionally high uplift and denudation rates at million-year time scales (King et al., 2016) and presently, 5 mm/yr and more (Zhao et al., 2023), and the Yarlung Tsangpo has an exceptionally high stream power, up to ~4000 W m$^{-2}$ in the Yarlung Tsangpo gorge into which the Sedongpu valley drains (Finnegan et al., 2008). At the Nyingchi meteorological station (c. 3000 m.asl., c. 50 km west of Sedongpu), the warmest months are July and August

(mean daily maximum around 20 °C, mean daily minimum 13 °C), and the coldest month is January (mean daily maximum 4 °C, mean daily minimum –8 °C). Highest mean monthly precipitations at Nyingchi are around 115 mm/month in July and August, and lowest in November and December with around 10 mm. Sedongpu Glacier rested on an elevated sediment/moraine bed (Haeberli et al., 2002) and the glacier surface had a c. 30-50 m higher elevation than the closest valley floor surrounding it. The wider study region shows the strongest glacier volume losses currently found in High-Mountain

Asia (Hugonnet et al., 2021). According to Obu et al. (2019), the Sedongpu Glacier was several hundred metres in elevation below the regional permafrost limit.

## 3 Data and Methods

Here, we mainly generate and investigate optical stereo data acquired by the Pleiades and Spot 6/7 satellites. We use the MicMac software (Rupnik et al., 2016; Rupnik et al., 2017) to produce DEMs and orthoimages of 13 Nov 2015 (Spot 6 tri-

stereo; before detachment), 30 Dec 2018 (Pleiades tri-stereo; two months after detachment), 12 Jan 2020 (Spot 7 stereo), 30 April 2021 (Pleiades tristereo), 19 Sep 2021 (Pleiades stereo), and 4 Nov 2022 (Spot 6 tristereo). All DEMs are generated and co-registered using standard procedures. Elevation and volume changes between our repeat DEMs can be estimated in different DEM combinations that should then sum up to the same amount. From the associated triangulation (or loop) errors we obtain an average uncertainty for our volume estimates of around $\pm 1 \cdot 10^6$ m$^3$. From stable ground areas we estimate a
standard deviation of elevations of $\pm 4$ m and a standard error of mean elevation changes over the glacier and rock avalanche sites of around $\pm 1$ m, translating to an uncertainty of around $\pm 5 \cdot 10^6$ m$^3$ for the volume change estimates. We choose the latter more conservative estimate as our volume change uncertainty.

## 4 The 2018 glacier detachment

In addition to a number of smaller mass flows from the catchment until 2017, two 17 and $33 \cdot 10^6$ m$^3$ rock-ice avalanches ran
from the Gyala Peri west flank over the Sedongpu Glacier late in October 2017 (Kääb et al., 2021; Li et al., 2022). Following the 2017 rock-ice avalanche(s), the lower part of Sedongpu Glacier underwent drastic changes. Ponds developed on its surface and along the margins, surface velocities increased from a background velocity of ~0.3 m day$^{-1}$ in 2017 to 25 m day$^{-1}$ in mid-October 2018 (Kääb et al., 2021; Zhang et al., 2022) and the glacier surface showed increased crevassing. In two parts on 16 Oct 2018 and 29 Oct 2018, the entire glacier tongue of in total $130 \pm 5 \cdot 10^6$ m$^3$ detached, below approximately
4050 m asl., complemented by an additional $\sim 44 \pm 5 \cdot 10^6$ m$^3$ from surrounding moraines (Figs. 1-2, Supplementary Fig. S3), leaving the sediment previously beneath the Sedongpu Glacier entirely exposed. Only the glacier parts in the steep headwall remained. The mass temporary dammed up the Yarlung Tsangpo (Liu et al., 2019; Tong et al., 2019; Wang et al., 2020; Kääb et al., 2021; An et al., 2022; Zhang et al., 2022). The theoretical ice thickness estimates for Sedongpu Glacier from Farinotti et al. (2019) agree on average well with the actual elevation loss between 2015 and December 2018 due to the
glacier detachment, indicating that the $130 \cdot 10^6$ m$^3$ glacier detachment volume might have consisted to a large extent of (though likely sediment-rich) ice rather than basal sediments. Pre-collapse ice-thickness datasets are however not of sufficient accuracy to evaluate whether the initial event was entirely composed of glacier ice, whether it entrained basal sediment, and what the volume of sediment entrained might have been. The ice thickness estimates by Millan et al. (2022) are roughly double and more of the above estimates and measurements, likely because they are based on 2017–2018 glacier
surface velocities, which were in the case of Sedongpu Glacier already increased due to pre-detachment surge-like acceleration (Kääb et al., 2021). In addition, relative velocity errors are particularly large for slow-flowing glaciers and these errors propagate into large uncertainties for the corresponding ice thickness reconstructions by Millan et al. (2022). This high ice content of the glacier detachment is confirmed by visual examination of the detachment zone and the deposits in post-collapse optical satellite data (Kääb et al., 2021) and the lack of a large erosional scar in the subglacial sediment, although we
could not confirm this by direct field observations.

## 5 Rapid and massive bed erosion

The most prominent elevation change between 2015 and December 2018 is the glacier detachment from October 2018 (Figs. 2, Tab. 1, Supplementary Fig. S3) (Kääb et al., 2021). Maximum detachment depths over the glacier area were c. 200 m. The total elevation changes from Dec 2018 (i.e. after glacier detachment) until 2022 show a massive erosion pattern over much of the former glacier bed and its surroundings, with maximum erosion depth of 360 m and an average of 135 m over an area of 2.5 km$^2$, amounting to about $335\pm5$ $10^6$ m$^3$. This volume corresponds to about 2.5 times the detached glacier volume (Figs. 1-2, Tab. 1, Supplementary Fig. S3). Between Dec 2018 and 2020, erosion depths of up to 30–50 m can be observed at limited areas along the deepest part of glacier detachment area. The elevation changes from January 2020 to April 2021 display a similar pattern. The April – September 2021 elevation changes constitute by far the largest part of the entire Dec 2018–2022 erosion of the glacier bed; $279\pm5$ $10^6$ m$^3$ with a maximum depth of around 310 m. During September 2021 and November 2022 another around 32 $10^6$ m$^3$ were eroded from the area of the newly formed canyon.

To better understand the massive erosion amounts detected between the 30 April and 19 September 2021 DEMs, we investigated optical and radar satellite data during this time period. Due to almost permanent cloud cover during this season over the study site, only very few useful optical data (Planet, Sentinel-2, Landsat) are available, and the most dense information stems thus from Sentinel-1 radar data. Combined, these data show a steady increase of the erosion area and suggest thus that the massive erosion happened gradually, or at least in a series of relatively small events, mainly during June–August and into early September 2021, rather than through one or a few massive landslides or debris flows. Between Dec 2018 (i.e. shortly after glacier detachment) and Apr 2021 the erosion was mainly concentrated along the drainage stream that developed through the detachment zone and corresponding avalanche path towards Yarlung Tsangpo (Fig. 2b, Suppl. Fig. S3). Suddenly in early June 2021, major erosion activity started at the point where the drainage stream left the former glacier bed (star in Fig. 1b and c, Fig. 2b). From this point in time and space, massive but gradual up-valley retrogressive erosion formed the main canyon until end of Aug 2021 (see Sentinel-1 images in Fig. 3 and the animation in the Supplement). In fact, state-of-the-art early warning installations including cameras and geophones at the outlet of the Sedongpu valley registered rock-ice avalanches (following section) but no massive debris flows from the former glacier bed and no river blockings of the Yarlung Tsangpo are reported (Yang et al., 2023). A new early warning station further up in the Sedongpu valley was only installed in May 2022, and was then also able to detect comparably smaller debris flows from the catchment. Assuming gradual, temporally constant erosion activity over June–August 2021 gives an extreme sediment flux of 3 $10^6$ m$^3$ every day over 3 months. Another indication that supports the interpretation of gradual erosion is the fact that the fan of the Sedongpu valley in the Yarlung Tsangpo showed rapid changes during summer 2021 but seemed to have never dammed up the main river (Sentinel-1 images in Fig. 3 and the animation in the Supplement) (Yang et al., 2023). Such damming happened after the 2018 glacier detachment. In theory, the sediments eroded in summer 2021 could have had a very high ice content, or consisting of dirty ice, so that the mass could quickly loose volume due to melt. We find however no indication of such ice in optical satellite data. The flanks of the erosion canyon are very steep, at most places more than

30° and many places even 40°–50°, suggesting the rapid summer 2021 erosion stopped for the time being at well-consolidated sediments.

## 6 Rock-ice avalanches and river erosion

Two other prominent landscape changes happened during 2018–2022 in the catchment. Further rock-ice volumes of in total $100\pm5$ $10^6$ m$^3$ were lost in the western flank of Gyala Peri, from the same area as the 50 $10^6$ m$^3$ rock-ice avalanche(s) of 2017, mentioned in Section 4. Of these roughly 100 $10^6$ m$^3$, 50 $10^6$ m$^3$ failed over Jan 2020–Apr 2021 (mostly on 22 March 2021; Bai et al. (2022); Zhao et al. (2022); Yang et al. (2023)) and another 50 $10^6$ m$^3$ during Apr 2021 – Nov 2022, all from the same area in the mountain flank (Fig. 1). These rock-ice avalanches are worth mentioning as their deposits, and the deposits of potential earlier avalanches, will have contributed to the large sediment volumes stored in the valley and have had impact on the ice properties of the Sedongpu Glacier, e.g. through particularly sediment-rich ice (Fig. 3f, Suppl. Fig. S3), and could also have directly affected the ice and sediment stability there, for instance the Sedongpu Glacier detachment. Second, our DEMs include also a reach of around 6 km downstream of the location where the Sedongpu valley joins the Yarlung Tsangpo. The fan of the Sedongpu valley shows a sequence of elevation gains and losses in response to the rock-ice mass flows from the Sedongpu valley, but there is no significant overall volume gain in the fan area and the ~6-km reach below (Fig. 2). This suggests that the river, having exceptionally high stream power (Section 2), was able to transport most of the massive amount of sediments that were deposited in particular during the 2018 glacier detachment and the June–September 2021 erosion peak. The Sentinel-1 image time series over summer 2021 (Fig. 3 and animation in the Supplement) shows rapid changes of the Sedongpu fan in extent, shape and height, but still these changes appear rather minor compared to the $279\pm5$ $10^6$ m$^3$ erosion volume that should have entered the fan during this time period. It would be interesting to investigate in a further study, whether and where signs of deposition of such large volume can be observed along the Yarlung Tsangpo and Brahmaputra (Zhao et al. (2022) mention increased river turbidity 200 km downstream after the March 2021 rock-ice avalanche).

## 7 Discussion and conclusions

In summary, between October 2017 and November 2022, around $659\pm7$ $10^6$ m$^3$ bedrock, sediment and ice debris have been exported from the Sedongpu catchment, most of it bedrock and sediments. About half of this volume ($335\pm5$ $10^6$ m$^3$) is estimated to be eroded from the bed of the former Sedongpu Glacier and its immediate surrounding. In the optical satellite images we find no indication of substantial amounts of massive ice in these sediments from the former glacier bed (An et al., 2022). This extreme erosion amount corresponds to erosion rates of up to 30 m per month during June–August 2021 for the erosion area itself, locally up to 100 m per month (Tab. 1), or up to two metres per month if calculated as a mean for the

entire basin area (total c. 65 km², c. 50 km² of it draining towards the Sedongpu Glacier). Such "ultra-rapid" rates of
paraglacial landscape response (Meigs et al., 2006) are to our best knowledge among the highest currently found on Earth.

An important question is to what extent general conclusions can be drawn from the extreme erosion volumes and rates that originate from the rapidly uncovered bed of Sedongpu Glacier. The subglacial material from below this glacier could be particularly prone to erosion. This would be in line with the assumption that large-volume detachments of low-angle valley glaciers seem to be associated with particularly soft glacier beds (Gilbert et al., 2018; Kääb et al., 2018; Kääb et al., 2021;
Leinss et al., 2021). Precipitation data from the Sedongpu catchment are not available to us and could in such extreme topography substantially differ from the measurements at Nyingchi, 50 km to the west. At Nyingchi, during June–mid September 2021, 28 days with precipitation amounts > 10 mm are recorded, six of which with 40–70 mm. These total and daily amounts seem to be substantially higher compared to the same time period in other years (Supplementary Figs. S1 and S2). The massive 2021 erosion could thus have been triggered by exceptionally high precipitation amounts and rates. The
main erosion activity, however, seems rather to have been a self-sustained, and perhaps self-enhancing, retrogressive destabilisation that, once triggered, formed the erosion canyon independent of precipitation amounts. This theory implies easily erodible, unconsolidated sediments, which are perhaps well water-saturated. The fact that the massive erosion activity in 2021 started exactly at the intersection of the former glacier boundary and the drainage stream (Sentinel-1 satellite images in Fig. 3 and animation in the Supplement) suggests that the former glacier bed was much more prone to erosion than the
surrounding moraines. Once a potential comparably stable protective surface layer on the former bed was incised through stream erosion, the underlying weak sediments could be mobilized. Alternatively, or additionally, enhanced stream erosion could have increased the terrain slope, or even undercut the former bed at the location of the erosion initiation. Such processes would not necessarily require any particularly high precipitation amounts and only be dependent on the time needed for the stream erosion to reach a (local) destabilisation threshold related to slope or spatial variation of sediment
properties. Precipitation data at the Nyingchi station, ERA5 reanalysis data, and GPM IMERG satellite-derived precipitation data all suggest no exceptionally large amounts of precipitation during the first half of August 2021. Still, Sentinel-1 radar data suggest continued massive erosion during that period (Fig. 3 and Supplementary animation). High precipitation amounts could however have cumulatively saturated the sediments to make them prone to destabilisation, or could have contributed to accelerated stream incision or critical increase in local terrain gradients. Numerical modelling of the landscape evolution at
Sedongpu could provide further constrains on the properties of the sediments and their mobilisation but is beyond the focus of this brief communication.

We have not systematically examined the erosion volumes after the one dozen other glacier detachments listed in Kääb et al. (2021). From visual interpretations of satellite imagery, though, we do not find as important extreme erosion in these other cases compared to Sedongpu, but note that some of the detachments are indeed associated with substantial post-detachment
erosion activity (e.g., Flat Creek (Jacquemart et al., 2022), Amney Machen, Rasht valley/Petra Pervogo range). Another special circumstance involved in the extreme erosion in the Sedongpu valley, in addition to the potentially pronounced soft sediments, could be the elevated glacier bed where particularly large amounts of sediments were stored underneath the

glacier, thus available to erosion. Sedimentary glacier beds should be widespread in most glacierized mountains on Earth (Benn and Evans, 1998). Even if not fully understood, they are a sign of an imbalance in sediment flux where the production exceeds the export capacity from a glacier catchment (Zemp et al., 2005) – not surprising for the comparably small Sedongpu catchment (total ca. 65 km², ca. 50 km² draining towards the glacier) that includes an enormous rock wall with substantial rock avalanche activity. Another open question is whether the erosion volumes at Sedongpu after rapid removal of the glacier are higher than they would be after gradual glacier retreat over decades and centuries. Some self-stabilizing effect could come into play over longer time intervals that does not act during rapid erosions, or vice-versa a self-enhancing process could act under such rapid erosions. At least over decadal to centennial time-scales, the recent events at Sedongpu Glacier seem to represent a rapid and irreversible process of landscape transformation from a sediment-filled glacier valley to a glacier-free one with a deeply incised canyon.

The wider implications of the massive 2018–2022 erosion from the Sedongpu basin for mountain landscape development and sediment fluxes depend on the spatial and temporal reference scales considered, including the significance of the event in the magnitude-frequency distribution of mountain sediment transport. Even compared to rates of pro- and post-glacial erosion that have so far been termed "ultra-rapid" (Meigs et al., 2006), the rates found in the Sedongpu valley since 2018 are exceptional. Compared to other glacier forefields, which typically show post-glacial erosion rates in the order of cm a$^{-1}$ (e.g., Delaney et al., 2018), the erosion volume at the former Sedongpu Glacier since 2018 is equivalent to several millennia of such average erosion rates. For entire glacier-hosting mountain ranges typical denudation rates are in the order of mm a$^{-1}$ (e.g., Islam et al., 1999; Hinderer et al., 2013; Thiede and Ehlers, 2013). Multiplying the Sedongpu Glacier catchment area (50 km²) by vertical motion rates of 5 mm yr$^{-1}$ (Zhao et al., 2023) gives an uplifted volume of 250 10$^6$ m$^3$ 1000 yr$^{-1}$ indicating that the rock and sediment volumes recently eroded from Sedongpu are roughly equivalent to the volumes uplifted over 1–2 millennia for the entire catchment, neglecting density differences. Distributing the 2018–2022 Sedongpu erosion volume to the entire area of the Brahmaputra catchment upstream of the location where the river leaves the Himalayan arch (Pasighat; ~250,000 km² catchment area) gives 1–2 mm (depending on whether the Sedongpu rock avalanches are included in the calculation or not). I.e., the recent Sedongpu erosion volume is by order of magnitude (compared to denudation rates from e.g. Islam et al., 1999; Thiede and Ehlers, 2013) equivalent to the annual denudation rate of the entire250,000 km² catchment of the Himalayan parts of the Brahmaputra. This implies that one or a few such events, in the present case triggered by glacier disappearance, can significantly vary the erosion rates of even one of the largest mountain river catchments on Earth.

The events at Sedongpu confirm that glaciers are able to protect their soft beds against massive erosion. Once uncovered, the erosion potential of soft glacier beds is here demonstrated to be possibly enormous for some glaciers in terms of volumes and rates. Such erosion could be particularly extreme under the existence of fine-grained subglacial sediments, and – possibly combined – for elevated glacier beds where especially large amounts of subglacial sediments are stored. The 2018–2022 landscape development at Sedongpu represents an extreme example of rapid post-glacial slope response, a process that is indeed expected to be potentially particularly strong close to the time of glacier loss (Knight and Harrison, 2014). Lake

outburst floods have been suggested to be major drivers of erosion in the Himalayas (Cook et al., 2018). The erosion volumes and rates at Sedongpu dwarf even those from lake outburst floods, though it is unclear how the frequency of both event types and thus their long-term volumes relate to each other. The changes at Sedongpu highlight an extreme glacier

erosion potential and hazards related to it from debris flows and impacts on rivers. For instance, only the exceptionally large sediment transport capacity of the Yarlung Tsangpo let the river accommodate the extreme short-term erosion volumes delivered to it without causing major river-damming. Such consequences of  shrinkage or disappearance of mountain glaciers have so far not been considered at this magnitude.

**Code availability**

The DEMs were generated using the open-source software MicMac (Rupnik et al., 2017; MicMac, 2022). The MicMac code used for the present study is available from Girod and Filhol (2022)

**Data availability**

Sentinel-1 and Sentinel-2 data are freely available from the ESA/EC Copernicus Sentinels Scientific Data Hub at https://scihub.copernicus.eu (Copernicus, 2020). Planet data (Dove and RapidEye) are not openly available as Planet is a

commercial company. However, scientific access schemes to these data exist (https://www.planet.com/markets/education-and-research/). The original Pléiades and Spot stereo images are commercial (Airbus) and under academic license not transferable to other users. Several derived products can be made available on request for academic use as defined under the Airbus license.

**Author contribution**

A.K. developed the study concept, wrote the text, did most analyses and prepared the figures. L.G. prepared the DEMs, and commented to and edited the text.

**Competing interests**

All authors declare that they have no competing interests.

**Acknowledgements**

We thank the editor Harry Zekollari, referee Max Van Wyk de Vries, and another anonymous referee for their detailed, thoughtful and constructive comments. This study was conducted under ESA Glaciers_CCI and EarthExplorer 10 Harmony projects, and the University of Oslo "EarthFlows" research initiative.

**Financial support**

This work was funded by the ESA project Glaciers_CCI (4000127593/19/I-NS), and the ESA EarthExplorer10 Mission
Advisory Group (4000127656/19/NL/FF/gp).

**Table 1: Volume changes in the Sedongpu (Tibet) catchment over 2015–2022 as a consequence of glacier detachment, glacier bed erosion and rock-ice avalanches.**

| DEM data | Volume change glacier location ($10^6$ m$^3$, M m$^3$, ± 5) | Maximum elevation change (m) | Average erosion rate (m a$^{-1}$) | Comment | Volume change rock-avalanche site ($10^6$ m$^3$, M m$^3$, ± 5) |
|---|---|---|---|---|---|
| 13 Nov 2015 | | | | | |
| | -174 | -200 | | Oct 2018 glacier detachment, c. 130 $10^6$ m$^3$ from glacier, rest surrounding moraine | -50 |
| 30 Dec 2018 | | | | | |
| | -12 | -50 | 6 | | -1 |
| 12 Jan 2020 | | | | | |
| | -12 | -70 | 6 | Contribution of rock-ice avalanche to changes at glacier location unclear | -50 |
| 30 Apr 2021 | | | | | |
| | -279 | -310 | 270 | Erosion mostly in June–August 2021 (c. 30 m month$^{-1}$) | -10 |
| 19 Sep 2021 | | | | | |
| | -32 | -120 | 10 | Contribution of rock-ice avalanche to changes at glacier location unclear | -40 |
| 4 Nov 2022 | | | | | |
| 12/2018-11/2022 | -335 | -360 | 30 | Maximum depths not at same location | -101 |
| 11/2015-11/2022 | -508 | -360 | 25 | Average erosion rate from just before glacier detachment to 11/2022 c. 45 m a$^{-1}$ | -151 |

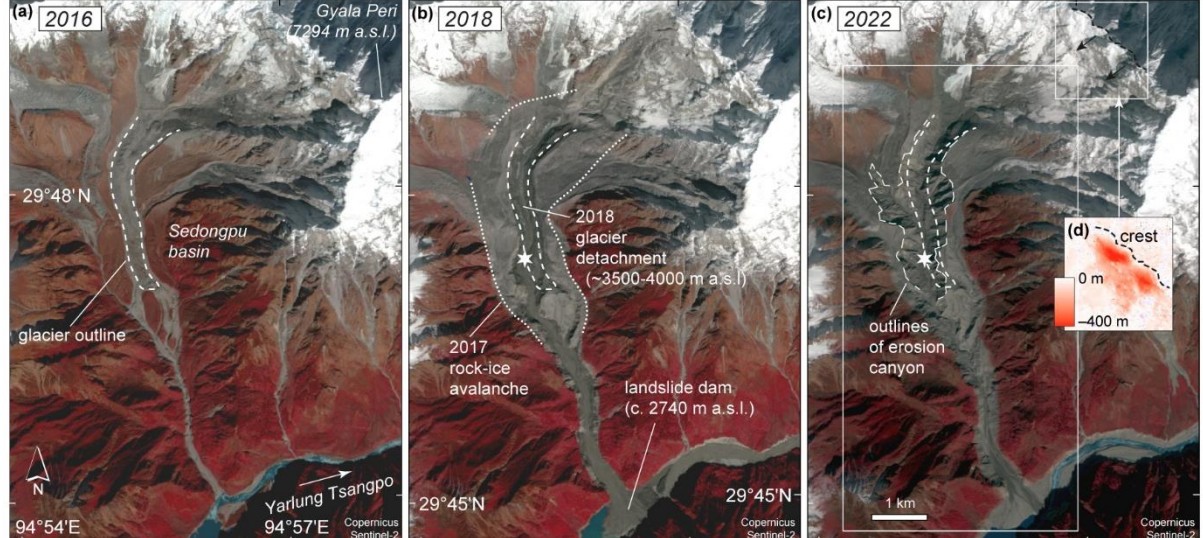

**Figure 1: Sedongpu basin on (a) 20 Nov 2016, (b) 31 Oct 2018, and (c) 19 Nov 2022 (credit: Copernicus Sentinel data). In 2017 a large rock-ice avalanche passed over the Sedongpu Glacier and in 2018, just before image (b) was taken, the glacier detached completely. Since then, a large ~335 $10^6$ m$^3$ canyon eroded at the location of the former glacier bed with by far highest erosion rates during summer 2021. The location where the massive 2021 erosion started is marked by a star. Inset (d): elevation changes 2015–2022 near the north crest of Gyala Peri showing the cumulative volume loss from three major ~50 $10^6$ m$^3$ rock-ice avalanches. The small white rectangle indicates the position of the elevation change of inset (d). The large white rectangle indicates the location of Fig. 2.**

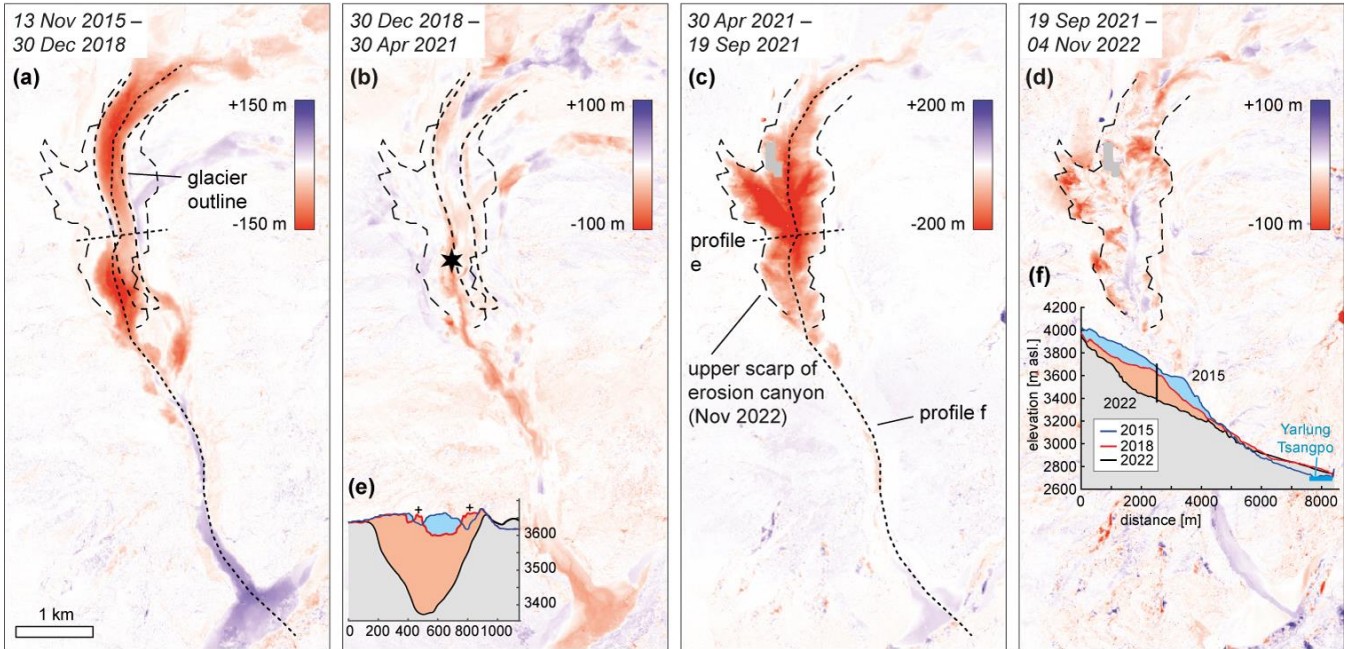

**Figure 2: Elevation changes over four periods (a)-(d) as derived from DEMs from optical satellite stereo data (Pleiades and Spot 6/7, stereo and tri-stereo). Note, the value ranges of the elevation change legends vary between panels, depending on the magnitude of observed changes. Corresponding elevation change rates are given in Table 1. For the star in panel (b) see Fig. 1. Insets (e) and**

**(f) are elevation profiles e and f of 13 Nov 2015, 30 Dec 2018 (after glacier detachment), and 4 Nov 2022. The + signs in inset (e) indicate 2015–2018 elevation increases from the 2017 rock-ice avalanche or splash deposits of the 2018 glacier detachment. For profile locations see panels (a) and (c). A larger version of insets (e) and (f) can be found in the Supplement.**


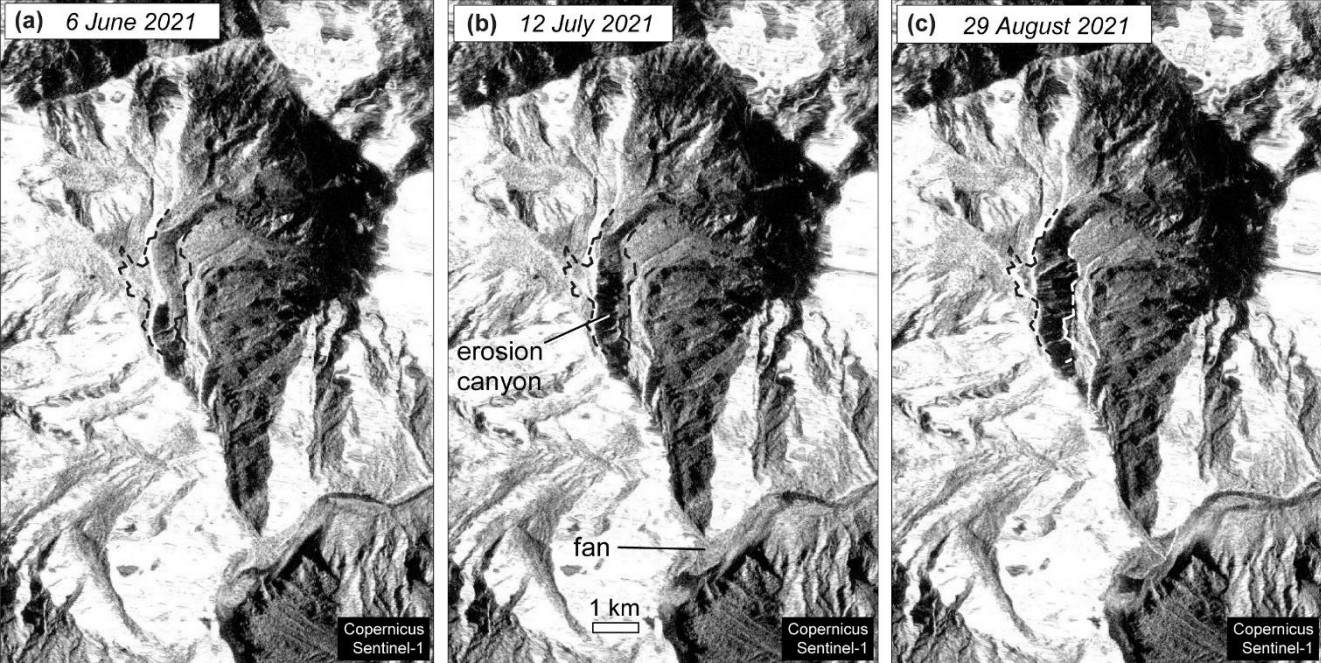

**Figure 3: Ortho-rectified Sentinel-1 images over Sedongpu (credit: Copernicus Sentinel data). The increasing dark area in the middle shows the evolution of the erosion canyon during summer 2021. Also note that the fan in the Yarlung Tsangpo is increasing in area. The lower-right margin of the fan appears bright in the image of 29.8.2021, indicating radar foreshortening and thus a**
**steeper front towards the river. For a full series of Sentinel-1 images see the gif-animation in the Supplement.**

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
