# Peer review of "Brief Communication: Rapid ~335 106 m3 bed erosion after detachment of the Sedongpu Glacier (Tibet)"

_The Cryosphere, 2022_

## Referee Comment (RC1)

TC-2022-237

Brief communication: Rapdi ~335 x $10^6$ m$^3$ bed erosion after detachment of the Sedongpu Glacier (Tibet)

Kääb & Girod
* * *
**General comments**

The authors have analyzed the changes of the Sedongpu valley following the 2018 glacier detachment. Given its incredibly high mass movement activity, the Sedongpu catchment is currently of high interest to the scientific community. In that sense, any contribution is of interest and additional data is surely welcome. While I think that the submitted manuscript is interesting, I do question its brevity and the would therefore like to suggest a few amendments.

I have two suggestions to increase the value of the presented contribution:
1) The authors initially make the connection with long-term landscape evolution, but later provide only one sentence where this topic is picked up again (L100). I would encourage a more detailed discussion of the topic. I am no expert at the topic, but a brief literature search reveals that the observed erosion rates at Sedongpu are far beyond the norm. Of course, long-term erosion rates are unlikely to be a good indicator of short-term peaks, but a more detailed literature review can surely shed more light on this than what I have done here. For example:
    a. Delaney et al. (2017) report an average of 5.4cm yr$^{-1}$ over 28 years at Griesgletscher, Switzerland
    https://onlinelibrary.wiley.com/doi/10.1002/esp.4239
    b. Hogan et al. (2020) report from values from Peterman Ice Stream that range from 0.5 mm yr$^{-1}$ to 1.5 mm yr$^{-1}$ during deglaciation cycles.
    https://tc.copernicus.org/articles/14/261/2020/tc-14-261-2020.pdf
    c. Hinderer et al. (2013) up to 7000 t km$^{-2}$ a$^{-1}$ from glacierized basins in the European Alps (5 to 10 times larger than non-glacierized basins)
    https://www.sciencedirect.com/science/article/pii/S0012825213000032?via%3Dihub
2) I very strongly encourage the authors to make their datasets open access to anyone. This would hugely increase the impact of this brief communication and surely provide the community with a really valuable dataset.

Specific comments:
L48: How far to the confluence with the Brahmaputra?
L40/41: *the* warmest months, *the* coldest moth
L44: How much do you think the elevated bed contributed to the overall erosion volume? Can you quantify this?
L47: *ran* not run

L52: This statement sounds a bit like the glacier slid off its sediment bed without any entrainment, which is likely not true. Can you quantify how much subglacial material was removed in the initial event (maybe using the ice thicknesses from Farinotti 2019 or Millan 2022?)

L66ff: Can you try to specify whether references to 2018 are before or after the detachment?

L78: *gradually,* not gradual

L81: *worth mentioning*

Table 1: Can you additionally express the changes in erosion rates (e.g. m yr$^{-1}$)? You say that the changes happened gradually, but it is hard to compare the different time periods because the time steps are very different. Would you expect some seasonality based on the strongly varying precipitation amounts? Does the data show this? If not, why not?

---

## Referee Comment (RC2)

Review of - *Brief communication: Rapid ~335 x 106 m³ bed erosion after detachment of the Sedongpu Glacier (Tibet)*

The authors have analysed a timeseries of digital elevation models to understand the post-detachment geomorphic change – specifically the formation of a deep canyon in the years following the rapid removal of Sedongpu glacier tongue. The paper is well written and the exceptional changes in this catchment will be of interest to the broader community. Some modifications to the paper in order to add a more in-depth exploration of some of the mechanisms driving the pulse of incision and the broader implications of these findings are needed. I believe that these changes can be made while remaining within the constraints of a 'Brief Communication'.

I have two major comments:
- The reason for the extremely rapid erosion between May-Sep 2021 needs to be explored in more detail. As it stands the manuscript clearly addresses the reasons for the rapid post-glacial erosion, but does not explain why the rates of erosion were ~an order of magnitude higher during one specific period, particularly a period not immediately following the detachment. This seems like one of the most important findings and one which may have wider implications for our understanding of these extreme erosive events. Why did the deposit experience a sudden change ~3 years post-detachment? The limited discussion about precipitation anomalies and other potential drivers needs to be expanded and moved to the methods/results instead of the discussion.

- The balance between results and discussions in this paper is heavily weighted towards results (L35-93). Even the discussions (L94-134) dedicate quite a lot of space to evaluating the specific findings of this study with a fairly limited consideration of the larger scale implications. There is a rich literature considering topics such as the evolution of post-glacial landscapes, the magnitude-frequency distribution of sediment transport events, the relative erosion rates of glacial and fluvial landscapes, and volumes of sediment removed from the Himalaya on different timescales. I would not expect the paper to discuss all possible implications in detail, but the impact of this paper will be substantially reduced if the (very important) findings are not more effectively placed within the context of the broader literature.

Minor comments line by line:

L6 "130 10⁶" – is there an uncertainty associated with this that can be quoted? There are a number of places throughout that volumes are given without uncertainties (e.g. L95, 96, etc), ideally these would all include an uncertainty from the discussed methods in L 60-65.

L7 remove 'drastic'

L10 Here, as a first-time reader, I am expecting to get some information about why there was extremely rapid erosion in 2021 specifically. I am left with a few questions in mind otherwise. The sentence "The mass was transported […]" on the other hand doesn't add very much.

L12 Ending the abstract with a hint about the broader implications of the work would be useful.

L14-34 This introduction is very well written. Concise, yet sets the scene well.

L25 I am not entirely convinced that a detachment can be considered a direct analogy of longer-timescale glacial retreat and landscape exposure as presented here. The rate of change matters – this detachment led to the instantaneous exposure of a very large area of unconsolidated sediment, which may have been more likely to partially stabilize if exposed gradually (e.g., through vegetation growth or other processes). A slight change in wording to get this across would be useful. You do discuss this later on L120 onwards.

L36-45 I am not sure this fits within the subheading '2018 glacier detachment' but is instead more general background information. Move to the intro or its own subheading?

L37 Is the Yarlung Tsangpo a 'tributary' to the Brahmaputra, or another name for the main branch of the same river?

L44 Reference for 'so called elevated bed'

L52 "leaving the bed of the Sedongpu glacier uncovered by ice" is confusing. If I understand correctly then the ice was gone, so the area is no longer the 'bed of the glacier'! Perhaps an alternate phrasing would be better, such as 'leaving the sediment previously beneath the Sedongpu glacier entirely exposed' or similar.

L56-64 Again, this first paragraph is not a great fit for the heading. This appears to be 'methods', not sure if it is best in its own heading or with the current heading modified to better represent it?

L75-79 Here is where details are seriously lacking about the extremely rapid erosion in summer 2021. The only real information about it here is "the massive erosion happened gradual, or at least in a series of smaller events". Please expand and discuss the potential drivers of this here (meteorological data or other) and the mechanisms by which the mass may have been removed (I am guessing some form of landslide/debris flow to evacuate that quantity of material in such a short time).

L81-85 Can you explain briefly why you consider this noteworthly? This is not immediately clear. Do you think that these subsequent avalanches played a role in destabilizing the sediment pile?

L87-88 Is this necessarily remarkable for a river of this scale? Also is there definitely no volume gain at all? It looks like a delta was constructed from the optical imagery in Fig 1.

L90-92 I can understand that an investigation of the routing of this sediment pulse be relegated to a 'further study', but this study will be easier to conduct if more baseline discussion about the sediment pulse is in this paper.

L95-96 Can you add the uncertainty to these numbers? Also, what is the distinction between 'rock' and 'sediment' – surely sediment is just a form of rock. Do you mean to distinguish between bedrock and sediment?

L98-99 'several tens to hundreds', 'few weeks to months', 'up to several metres' – this sentence has a lot of vague terms. Can you just replace with the actual exact numbers which you have in the results above? They will be more informative.

L102-103 People have mixed views about rhetorical questions, I am OK with them in general. However, I am not sure you really respond to this in any detail in the rest of the paragraph. Perhaps remove it, or keep it in an expanded discussions section about the broader implications (that is needed in any case).

L103-104 "The subglacial material from below this glacier seems especially easy eroded." As far as I can tell you haven't really presented the data to support this. We know that is was eroded, but if the erosion was driven by an extreme rainstorm or similar then it may not necessarily be particularly easily erodible?

L106-110 This exploration of possible causes is too brief. It might be useful to have a plot subpanel showing the precipitation data, and possibly exploring reanalysis (MSWEP/ERA5-land/CHIRPS) or satellite-derived (GPM IMERG) precipitation over the area of interest would be useful. These have low spatial resolution (0.05-0.1 degree) and have errors in areas like this, but may be useful for revealing relative patterns.

L117 "Such elevated glacier beds are widespread in most glacierized mountains on Earth" should have a reference.

L126 remove 'impressively'

L126-132 There are hints of discussion of the results in a broader context, but the broader literature is mostly absent. Please add in some external context to better highlight why these results are particularly relevant.

Data availability – It is a shame that the data cannot be shared more easily, but I understand that this is a limitation of these commercial datasets.

Fig 1 and 2 – the meanings of the black and white dashed lines on the maps are not easy to find. Could you add some info about these to the caption? Also, the white boxed in Fig 1 are not easy to see, perhaps another colour would contrast better with the ice?

Fig 2 – I am not sure how this may best be integrated into the current figure, but the erosion would be more intuitive and easier to compare as a rate (m/yr of erosion for example) than simply an elevation loss. Could this be overlaid onto the existing color legend?

-Max Van Wyk de Vries

---

## Author Comment (AC1)

TC-2022-237
**Brief Communication: Rapid ~335 $10^6$ m$^3$ bed erosion after detachment of the Sedongpu Glacier (Tibet)**
Andreas Kääb, Luc Girod

**Revisions in response to referees**

**General response**

We would like to thank the two referees for their thoughtful and constructive reviews that certainly helped to improve the paper. We implemented all recommended changes and in particular added information and discussion about the concentration of erosion in summer 2021 and about the general implications of the events with respect to landscape evolution.

Editor/referee comments are in *italic*, and our response in normal font.

An annotated version of our revised manuscript with track changes is attached.

**Response to referee #1 ………….. page 1**

**Response to referee #2 ………….. page 4**

**Response to Referee #1, anonymous**

*General comments*

*The authors have analyzed the changes of the Sedongpu valley following the 2018 glacier detachment. Given its incredibly high mass movement activity, the Sedongpu catchment is currently of high interest to the scientific community. In that sense, any contribution is of interest and additional data is surely welcome. While I think that the submitted manuscript is interesting, I do question its brevity and the would therefore like to suggest a few amendments.*

*I have two suggestions to increase the value of the presented contribution:*

*The authors initially make the connection with long-term landscape evolution, but later provide only one sentence where this topic is picked up again (L100). I would encourage a more detailed discussion of the topic. I am no expert at the topic, but a brief literature search reveals that the observed erosion rates at Sedongpu are far beyond the norm. Of course, long-term erosion rates are unlikely to be a good indicator of short-term peaks, but a more detailed literature review can surely shed more light on this than what I have done here. For example:*

*Delaney et al. (2017) report an average of 5.4cm yr$^{-1}$ over 28 years at Griesgletscher, Switzerland https://onlinelibrary.wiley.com/doi/10.1002/esp.4239*

*Hogan et al. (2020) report from values from Peterman Ice Stream that range from 0.5 mm yr$^{-1}$ to 1.5 mm yr$^{-1}$ during deglaciation cycles. https://tc.copernicus.org/articles/14/261/2020/tc-14-261-2020.pdf*

*Hinderer et al. (2013) up to 7000 t km$^{-2}$ a$^{-1}$ from glacierized basins in the European Alps (5 to 10 times larger than non-glacierized basins) https://www.sciencedirect.com/science/article/pii/S0012825213000032?via*

*%3Dihub*

We modified text at several places and added a paragraph in the discussion section, see more details under major comment 2) of referee #2.

*I very strongly encourage the authors to make their datasets open access to anyone. This would hugely increase the impact of this brief communication and surely provide the community with a really valuable dataset.*

We will try our best without breaching the strict licensing conditions. We are definitely not allowed to make our DEMs open access, but could make dDEMs available for academic purpose. This is why we need to ask for requests.

*Specific comments:*

*L48: How far to the confluence with the Brahmaputra?*

Clarified that "Yarlung Tsangpo" is actually Tibetan for "Brahmaputra". Our study site is at the main branch of Brahmaputra even if Brahmaputra has a number of large other tributaries. See also response to ref#2 L37.

*L40/41: the warmest months, the coldest moth*

Done

*L44: How much do you think the elevated bed contributed to the overall erosion volume? Can you quantify this?*

See L52 comment below

*L47: ran not run*

Done

*L52: This statement sounds a bit like the glacier slid off its sediment bed without any entrainment, which is likely not true. Can you quantify how much subglacial material was removed in the initial event (maybe using the ice thicknesses from Farinotti 2019 or Millan 2022?)*

We added

"The theoretical ice thickness estimates for Sedongpu Glacier from Farinotti et al. (2019) agree on average well with the actual elevation loss 2015 – December 2018 due to the glacier detachment, suggesting that the $130 \cdot 10^6$ m$^3$ glacier detachment volume consisted to a large extent of (though likely sediment-rich) ice rather than basal sediments. This high ice content of the glacier detachment is confirmed by visual interpretation of the deposits (Kääb et al., 2021). The ice thickness estimates by Millan et al. (2022) are roughly double and more of the above estimates and measurements, likely because they are based on 2017–2018 glacier surface velocities, which were in the case of Sedongpu Glacier already elevated due to pre-detachment surge-like acceleration (Kääb et al., 2021)."

*L66ff: Can you try to specify whether references to 2018 are before or after the detachment?*

Good catch! Clarified.

*L78: gradually, not gradual*

Done

*L81: worth mentioning*

Done

*Table 1: Can you additionally express the changes in erosion rates (e.g. m yr$^{-1}$)? You say that the changes happened gradually, but it is hard to compare the different time periods because the time steps are very different.*

Rough erosion rates included in the table.

*Would you expect some seasonality based on the strongly varying precipitation amounts? Does the data show this? If not, why not?*

One would expect that the erosion has some seasonal dependence with precipitation amounts, but the temporal sampling of our DEMs is too coarse for such conclusions. In fact, the most massive erosion in 2021 starts in June, during the rain season. As we explain below in response to referee #2, this erosion start is, however, not necessarily due to large precipitation amounts.

*The authors have analysed a timeseries of digital elevation models to understand the post-detachment geomorphic change – specifically the formation of a deep canyon in the years following the rapid removal of Sedongpu glacier tongue. The paper is well written and the exceptional changes in this catchment will be of interest to the broader community. Some modifications to the paper in order to add a more in-depth exploration of some of the mechanisms driving the pulse of incision and the broader implications of these findings are needed. I believe that these changes can be made while remaining within the constraints of a 'Brief Communication'.*

*I have two major comments:*

*1) The reason for the extremely rapid erosion between May-Sep 2021 needs to be explored in more detail. As it stands the manuscript clearly addresses the reasons for the rapid post- glacial erosion, but does not explain why the rates of erosion were ~an order of magnitude higher during one specific period, particularly a period not immediately following the detachment. This seems like one of the most important findings and one which may have wider implications for our understanding of these extreme erosive events. Why did the deposit experience a sudden change ~3 years post-detachment? The limited discussion about precipitation anomalies and other potential drivers needs to be expanded and moved to the methods/results instead of the discussion.*

We added new information to the results section and additional discussion:

"Between Dec 2018 (i.e. shortly after glacier detachment) and Apr 2021 the erosion was mainly concentrated along the drainage stream that developed through the detachment area and corresponding avalanche path towards Yarlung Tsamgpo (Fig. 2b). Suddenly in early June 2021, major erosion activity started at the point where the drainage stream left the former glacier bed (star in Fig. 1b and c, Fig. 2b). From this point in time and space, massive but gradual up-valley retrogressive erosion formed the main canyon until end of Aug 2021. Assuming gradual constant erosion activity over June–August 2021 gives an extreme sediment flux of 3 $10^6$ m$^3$ every day over 3 months."

And

"The main erosion activity seems however rather to have been an autonomous, and perhaps self-enhancing, retrogressive destabilisation that, once triggered, formed the erosion canyon independent of precipitation amounts. This theory implies easily erodible, unconsolidated sediments, perhaps well water-saturated. The fact that the massive erosion activity in 2021 started exactly at the intersection of the former glacier boundary and the drainage stream suggests that the former glacier bed was much more easily eroded than the surrounding moraines. Once a comparably stable potential protective surface layer on the former bed was incised through stream erosion, the underlying weak sediments

could be mobilized. Alternatively, or additionally, enhanced stream erosion could have increased the terrain slope, or even undercut, at the location of the erosion initiation. Such processes would not necessarily require any particularly high precipitation amounts and only be dependent on the time needed for the stream erosion to reach a (local) destabilisation threshold related to slope or spatial variation of sediment properties. Precipitation data at the Nyingchi station, ERA5 reanalysis data, and GPM IMERG satellite-derived precipitation data all suggest very little precipitation during the first half of August 2021. Still, Sentinel-1 radar data suggest continued massive erosion during that period. High precipitation amounts could have particularly saturated the sediments to make them prone to destabilisation, or could have contributed to accelerated stream incision or critical increase in local terrain gradients, though."

We further add now a Supplement to the paper containing precipitation data and the animation of repeat Sentinel-1 data, illustrating the gradual erosion during summer 2021.

*2) The balance between results and discussions in this paper is heavily weighted towards results (L35-93). Even the discussions (L94-134) dedicate quite a lot of space to evaluating the specific findings of this study with a fairly limited consideration of the larger scale implications. There is a rich literature considering topics such as the evolution of post-glacial landscapes, the magnitude-frequency distribution of sediment transport events, the relative erosion rates of glacial and fluvial landscapes, and volumes of sediment removed from the Himalaya on different timescales. I would not expect the paper to discuss all possible implications in detail, but the impact of this paper will be substantially reduced if the (very important) findings are not more effectively placed within the context of the broader literature.*

We modified the text at several places and added a paragraph in the discussion section:

"The wider implications of the massive 2018–2022 erosion from the Sedonpgu basin for mountain landscape development and sediment fluxes depend on the spatial and temporal reference scales considered, including the significance of the event in the magnitude-frequency distribution of mountain sediment transport. Even compared to rates of pro- and post-glacial erosion that have so far been termed "ultra-rapid" (Meigs et al., 2006) the rates found in the Sedongpu valley since 2018 are exceptional. Compared to other glacier forefields which typically show post-glacial erosion rates in the order of cm a$^{-1}$ (e.g., Delaney et al., 2018), the erosion volume at the former Sedongpu Glacier since 2018 is equivalent to several millennia of such average erosion rates. For entire mountain ranges hosting glaciers typical denudation rates are in the order of mm a$^{-1}$ (e.g., Gabet et al., 2008; Hinderer et al., 2013; Thiede and Ehlers, 2013). Distributing the 2018–2022 Sedongpu erosion volume to the entire area of the Brahmaputra catchment upstream of the location where the river leaves the Himalayan arch (Pasighat) gives 1–2 mm (depending on whether the rock avalanches are included or not), i.e. the recent Sedongpu erosion volume is by order of magnitude equivalent to the annual denudation rate of a ~250,000 km$^2$ catchment. This implies that one or a few such events, triggered by glacier disappearance, can significantly vary the erosion rates of even one of the largest mountain river catchments on Earth."

*Minor comments line by line:*

*L6 "130 10$^6$" – is there an uncertainty associated with this that can be quoted? There are a number of places throughout that volumes are given without uncertainties (e.g. L95, 96, etc), ideally these would all include an uncertainty from the discussed methods in L 60-65.*

Added uncertainties at a number of places. In addition we managed now to process the 2022 DEM at the same accuracy as the other DEMs. The respective explanation for the 2022 DEM was removed.

*L7 remove 'drastic'*

Done

*L10 Here, as a first-time reader, I am expecting to get some information about why there was extremely rapid erosion in 2021 specifically. I am left with a few questions in mind otherwise. The sentence "The mass was transported […]" on the other hand doesn't add very much.*

We modified the abstract and added that the summer 2021 erosion peak was through temporally concentrated but still gradual retrogressive erosion into the former glacier bed, or a series of "smaller" events. (More details above and under L75-79 comment).

*L12 Ending the abstract with a hint about the broader implications of the work would be useful.*

Done

*L14-34 This introduction is very well written. Concise, yet sets the scene well.*

Thanks

*L25 I am not entirely convinced that a detachment can be considered a direct analogy of longer-timescale glacial retreat and landscape exposure as presented here. The rate of change matters – this detachment led to the instantaneous exposure of a very large area of unconsolidated sediment, which may have been more likely to partially stabilize if exposed gradually (e.g., through vegetation growth or other processes). A slight change in wording to get this across would be useful. You do discuss this later on L120 onwards.*

We term the analogy now "indication of the maximum erosion potential that might else be mobilized over longer time scales of gradual glacier retreat".

*L36-45 I am not sure this fits within the subheading '2018 glacier detachment' but is instead more general background information. Move to the intro or its own subheading?*

Agreed. We put this information in a new section "Study site".

*L37 Is the Yarlung Tsangpo a 'tributary' to the Brahmaputra, or another name for the main branch of the same river?*

To our best knowledge there are different perceptions around. We use now the one used by among others the Encyclopedia Britannica, i.e. Yarlung Tsangpo (Tibetan name) for the upper (=Tibetan) reaches of Brahmaputra.

*L44 Reference for 'so called elevated bed'*

Added a reference

*L52 "leaving the bed of the Sedongpu glacier uncovered by ice" is confusing. If I understand correctly then the ice was gone, so the area is no longer the 'bed of the glacier'! Perhaps an alternate phrasing would be better, such as 'leaving the sediment previously beneath the Sedongpu glacier entirely exposed' or similar.*

Changed

*L56-64 Again, this first paragraph is not a great fit for the heading. This appears to be 'methods', not sure if it is best in its own heading or with the current heading modified to better represent it?*

We put this information in a new section "Data and Methods".

*L75-79 Here is where details are seriously lacking about the extremely rapid erosion in summer 2021. The only real information about it here is "the massive erosion happened gradual, or at least in a series of smaller events". Please expand and discuss the potential drivers of this here (meteorological data or other) and the mechanisms by which the mass may have been removed (I am guessing some form of landslide/debris flow to evacuate that quantity of material in such a short time).*

See response to above major comment. We added according explanation to the text.

*L81-85 Can you explain briefly why you consider this noteworthy? This is not immediately clear. Do you think that these subsequent avalanches played a role in destabilizing the sediment pile?*

Modified to read "These rock-ice avalanches are worth mentioning as their deposits will have contributed to the ice and sediment properties in the valley, and could have also directly affected the ice and sediment stability there, for instance the Sedongpu Glacier detachment."

*L87-88 Is this necessarily remarkable for a river of this scale? Also is there definitely no volume gain at all? It looks like a delta was constructed from the optical imagery in Fig 1.*

We removed "remarkably", but still think this is important information that we would like to keep. Between some individual DEMs there are of course significant elevation changes in the river bed but they seem to be removed in sum within quite short time periods (few years maximum). The new delta volume is largely balanced by strong river erosion along the Yarlung Tsangpo just below the delta (likely due to changed river course) rendering the overall mass balance over and around the delta area quite small compared to the massive imported volumes.

*L90-92 I can understand that an investigation of the routing of this sediment pulse be relegated to a 'further study', but this study will be easier to conduct if more baseline discussion about the sediment pulse is in this paper.*

Done. See response to above major comment.

*L95-96 Can you add the uncertainty to these numbers? Also, what is the distinction between 'rock' and 'sediment' – surely sediment is just a form of rock. Do you mean to distinguish between bedrock and sediment?*

Done

*L98-99 'several tens to hundreds', 'few weeks to months', 'up to several metres' – this sentence has a lot of vague terms. Can you just replace with the actual exact numbers which you have in the results above? They will be more informative.*

Done

*L102-103 People have mixed views about rhetorical questions, I am OK with them in general. However, I am not sure you really respond to this in any detail in the rest of the paragraph. Perhaps remove it, or keep it in an expanded discussions section about the broader implications (that is needed in any case).*

Sentence modified and discussion added in response to above major comment.

*L103-104 "The subglacial material from below this glacier seems especially easy eroded." As far as I can tell you haven't really presented the data to support this. We know that is was eroded, but if the erosion was driven by an extreme rainstorm or similar then it may not necessarily be particularly easily erodible?*

We modified the sentence to list soft sediments as one possible cause of the rapid erosion. We also added discussion about the temporal concentration and cause of the erosion in response to above major comment. This discussion points also to soft (or else highly erodible) sediments.

*L106-110 This exploration of possible causes is too brief. It might be useful to have a plot subpanel showing the precipitation data, and possibly exploring reanalysis (MSWEP/ERA5-land/CHIRPS) or satellite-derived (GPM IMERG) precipitation over the area of interest would be useful. These have low spatial resolution (0.05-0.1 degree) and have errors in areas like this, but may be useful for revealing relative patterns.*

We add now a Supplement to the paper, containing meteorological data (ERA5 and station data).

*L117 "Such elevated glacier beds are widespread in most glacierized mountains on Earth" should have a reference.*

We added a reference to elevated sediment beds in the site description. We modified the sentence to refer to sedimentary beds in general, using Zemp et al 2005 as reference.

*L126 remove 'impressively'*

Done

*L126-132 There are hints of discussion of the results in a broader context, but the broader literature is mostly absent. Please add in some external context to better highlight why these results are particularly relevant.*

See response to major comment 2)

*Data availability – It is a shame that the data cannot be shared more easily, but I understand that this is a limitation of these commercial datasets.*

We will try our best without breaching the strict licensing conditions. We are definitely not allowed to make our DEMs open access, but could make dDEMs available for academic purpose. This is why we need to ask for requests.

*Fig 1 and 2 – the meanings of the black and white dashed lines on the maps are not easy to find. Could you add some info about these to the caption? Also, the white boxed in Fig 1 are not easy to see, perhaps another colour would contrast better with the ice?*

Modified in particular Fig 1 and added explanations in the figure.

*Fig 2 – I am not sure how this may best be integrated into the current figure, but the erosion would be more intuitive and easier to compare as a rate (m/yr of erosion for example) than simply an elevation loss. Could this be overlaid onto the existing color legend?*

We are unsure if showing erosion rates (instead of absolute changes or in addition) works well. We admit we haven't found a completely satisfying solution. The problem is that the rates are randomly dependent on the times of DEMs. For instance for the 2021 erosions, should we use the DEM dates as reference time or the shorter time period where the erosion actually happened? We believe showing the measured elevation changes is the most direct and re-usable result, free of hypotheses about the rate reference time. We added erosion rates now in Table 1, believing that the table offers more flexibility to express these erosion rates than the figure. We referred now in the figure caption to the table.

---

## Referee Report (RR1)

Review of author response- *Brief communication: Rapid ~335 x 10$^6$ m$^3$ bed erosion after detachment of the Sedongpu Glacier (Tibet)*

I thank the authors for their rapid response to the two reviews and edits to the manuscript. The changes to the manuscript go a long way towards addressing my previous concerns, particularly by enhancing the discussion around the causes of the extremely rapid erosion episode and the broader implications of the work. I am glad to see this newly added sentence in the abstract "*The recent erosion volumes at Sedongpu are by order of magnitude equivalent to the average annual denudation volume of the entire mountainous part of the Brahmaputra River basin, and illustrate a potential and intensity for rapid post-glacial landscape evolution and the hazards related to such high-magnitude low-frequency events that have rarely been considered so far.*" which I think will interest a whole new group of potential readers.

Overall, I recommend the authors make minor revisions to complete a few of the additional changes described here, after which this manuscript would be suitable for publication in TC.

If the intention was to create a full-length manuscript, I would have recommended that the authors add in a component of landscape modelling to the manuscript alongside the remote sensing. The two methods would complement each other nicely in evaluating this extreme event, and may allow for some constrains on the properties of the sediment (e.g. erodibility). This remote-sensing only manuscript makes for a good Brief Communication with the changes and does not have space for this added material, but it may be worth noting briefly in the discussions. It could make for a good follow-up paper to this.

Finally, the new supplementary material is very useful and should be discussed in a little more detail. The Sentinel-1 timeseries shows the rapid unzipping of the landscape in a way that is not currently captured in the manuscript. In addition, the period of rapid change in the delta within the Tsangpo from June to Aug 2021 provides some clues into when sediment was being delivered to this river. I would like to see 1-2 more sentences describing this. I am not sure if there is space for a new figure in the manuscript, but showing the following three images really highlights the processes occurring during the period between the two DEMs (e.g. below). Maybe it could be a figure in the sup mat and referred to directly.

[Figure]

Finally, I am wondering about one other potential implication of this event. The volumes of sediment mobilized are on the same order as a very large landslide. This sediment happened to be delivered to the Tsangpo, one of the rivers with the greatest sediment transport capacity in the globe, so could largely be accommodated into the system. However, if this had occurred in a smaller catchment, there would be a very high chance of the river being temporarily dammed with associated outburst flood risk. It is somewhat speculative, but it could be useful to note this point in the broader implications.

 A few specific points:

L8 (and elsewhere) remove 'River', it is not needed.

L15 'mountainous part of the…' Not quite clear what this means. Could you be more clear and reword?

L42 maybe "Yarlung Tsangpo (also known as the Brahmaputra in its lower reaches)"?

L52 'should have been' -> 'was'

L73-80 I understand that this material was added in response to the other review's question about removal of material in the initial event, but I am not very convinced by it. The ongoing destabilization of this glacier raises questions about many of the assumptions underpinning the ice-thickness calculations in both the Farinotti et al., 2019 and Millan et al., (2022) datasets. The problems may be more apparent in Millan et al.'s dataset, but the Farinotti et al., dataset may match the elevation loss by coincidence (examining the spread within the different models averaged at this location may give some idea). Finally, the uncertainties in both of these datasets for an individual glacier are much larger than the DoD and I am not sure about the usefulness of this comparison. You can mention it, but it will need to be framed by more discussion about the inherent uncertainties of these data.

Beyond this, I am not sure how much it matters whether the initial collapse was entirely composed of ice or entrained basal sediment for the remainder of this manuscript. If you say something along the lines of 'Pre-collapse ice-thickness datasets are not of sufficient accuracy to evaluate whether the initial event was entirely composed of glacier ice, whether it entrained basal sediment, and what the volume of sediment entrained might have been. Examination of post-collapse optical imagery could not identify a large erosional scar in the subglacial sediment (Kaab et al., 2021), although this was not confirmed by direct field observations.'

L103 I am genuinely astonished that this volume of material could be removed without the occurrence of debris flows. I only had time to have a very quick read through Yang et al.'s preprint, but do you have an idea what scale of debris flow could have been missed by their equipment? It sounds like it was moved to a point higher in the channel in 2021 following the March event, so may have been less sensitive?

L143-150 It would be good to refer to the Sentinel-1 imagery in this, as it supports the description (which appears a little speculative without it).

L180-183 Again, really happy to see this larger-scale description here, which is one of the most remarkable findings in my view. This sentence needs a reference (or several) for the source of the basin-wide erosion rate data.

L192-193 The two halves of this sentence are not entirely equivalent. While the volume and rate of the Sedomgpu erosion dwarf GLOF, the relative frequency of each is (as far as I know) not known. This should be added to the sentence or reworded so it is not implying that these events are an even larger driver of erosion in the Himalaya (which I don't think is what you are trying to say).

Code availability: This change is good and it makes it easier for readers to find the exact information.

-Max Van Wyk de Vries

---

## Editor Decision (ED1)

**Editor report for brief communication 'Rapid ~335 106 m3 bed erosion after detachment of the Sedongpu Glacier (Tibet)' by Kääb and Girod**
**May 2023**

Dear Andreas Kääb and Luc Girod,

Thanks a lot for having responded to the new comments during the second round of revisions and for having updated the manuscript accordingly. When comparing the current manuscript to the version at the stage of initial submission (December 2022), I think it is fair to say that the manuscript has improved in clarity and that the additional details that you added make the story very interesting, also for those who are not directly in the field of glacier geomorphology/erosion/hazards (like me). I am convinced that this short ,yet very clear story will be of interest to the readers of *The Cryosphere*. After reading the latest version of your manuscript, I have formulated a series of mostly minor and easy to incorporate suggestions that I hope you will find helpful. I invite you to consider these comments, after which we should normally be able to proceed to the final acceptance of your manuscript.

- l.12: probably best to have a consistent use of $m^3$ or $km^3$ to make quantities directly comparable. Not only here, but throughout the manuscript.
- l.15-16: this last sentence of the abstract was quite difficult to understand. Consider rewriting to something along the lines of: "…the Himalayas. This high-magnitude low-frequency event illustrates a potential for rapid post-glacial landscape evolution and associated hazards that have rarely been observed (at such high intensity) so far".
- l.19: "…disappearance, these newly uncovered areas are… "
- l.23: "…comparably slowly, over…"
- l.26: not sure you need "respectively" here, since you do not refer to anything mentioned before in a given order. Would suggest removing this here. Same for occurrence on l. 67.
- l.29: "…indication on the maximum…"
- l.30: "…detachment, entire…" (also other occurrences where a "," would be needed: e.g., in l. 53 "Obu et al. (2019), …"), l.99 ("..study site, only very…")
- l.36-38: hard to understand. Possibly change to: "We summarize key site information on the 2018 glacier detachment, and quantify the glacier-bed volume changes and other landscape changes in the basin until 2022" (possibly even add until which month in 2022)
- l.40: for the study site description, in the current formulation, it seems like there is no glacier remaining at all? While in reality a part of the glacier survived / did not collapse? Would be good to specify this a bit more. Also, to frame it better, maybe start the sentence with: "At the time of its detachment, the Sedongpu glacier was…"
- l.40: elevation of about 3700 m: could you provide the elevation range of the glacier at the time of detachment? And possibly also for what is now remaining of the glacier?
- l.42: "… The highest point"
- l.43-44: extreme angles of the slopes: could you provide a quantification for this statement? What slope for the angles are we to expect here?
- l.45: "…Tsangpo has an…"
- l.52: "The wider study region…"

- l.68: "…avalanches ran from the Gyala west flank over…"
- l.72: "entire tongue": so from this I tend to understand that the entire glacier did not collapse? See related comment above. Would be good to have a quantitative indication about how much of the glacier was lost and e.g., the elevation range of the glacier before and after the collapse.
- l.72: possibly reword to: "…detached, complemented by an additional…"
- l.82: yes, indeed quite high uncertainties for the ice thickness reconstruction. Aside from the change in velocity, the fact that relative errors are very large for velocities of slow-flowing glaciers also leads to a large (relative) error in the corresponding ice thickness reconstruction by Millan et al. (2022). Would be good to mention this in one or two additional sentences.
- l.89: "until 2022" (add white space)
- l.88-92: quite a long and fragmented sentence. Suggest splitting this up in two sentences, e.g., "and its surroundings, with maximum erosion depth of 360 m and an average of 135 m over an area of 2.5 km$^2$, amounting to about 335+-5 $10^6$ m$^3$. This volume corresponds to about 2.5 times the detached glacier volume (Figs. 1-2…"
- l.92: "…can be observed at limited…"
- l.93: "…elevation changes from January…"
- l.94: "…contribute by far to the largest…"
- l.110-111: "… (Yang et al., 2023). A new early…May 2022, and was then also…"
- l.118: glacier bed being "likely temperate": is there any evidence for this statement? Measurements and/or modelling of glaciers in this region? Would be good to specify and provide additional info for this.
- l.127: "contributed to the ice and sediment properties in the valley": sounds a bit vague/mysterious here: can this be reformulated to be more specific? Or possibly remove this? (the sentence also works well without this)
- l.132: maybe reword to "…was able to transport most of the…": i.e. omit "further"
- l.136: "It would be interesting…"?
- l.141: "In summary, between early 2017 and November 2022, around…": and ideally, be even more specific for what early 2017 is (i.e., which month)
- l.141: 659 +- 7: this +-7 remains a remarkably small error estimate (i.e., a mere 1% of the total volume)…
- l.141-143: suggest splitting up in two separate sentences: "…bedrock and sediments. About half of time volume (335+-5 $10^6$ m$^3$) is estimated to be eroded from the…"
- l.144: "…in the latter volume": what is this exactly? Can you be more specific here?
- l.151: "…could be particularly prone to erosion. This…" + on l.162: "…bed was much more prone to erosion than the…"
- l.160: "…sediments, which are perhaps…"
- l.169: very little precipitation. Has this been quantified, and could you provide a figure for these numbers? e.g., how this compared to other (standard) years, with this year having for instance X% less precipitation?
- l.172: "…terrain gradients. Numerical modelling": i.e., suggest removing the "though" here.
- l.174: other glacier detachments. Can you mention here how many detachments these are? e.g., "…detachments (X in total) listed in Kääb et al. (2021)"

- l.175: suggest rewording to: "…, we do not find as important extreme erosion in these other cases compared to Sedongpu, but…"
- l.178: "…potentially pronounced soft sediments…"
- l.180: "…most glacierized mountains on Earth": reference for this statement?
- l.200: when calculating the size of the hypothetical catchment, you may want to refer to how much larger this is than the actual catchment, e.g., "…catchment (X times more than actual size of the catchment)"
- l.211: unclear what signal is in change of GLOFs. Would be worth mentioning recent study in Nature by Veh et al. (2023), who suggest that GLOFs are reducing in frequency (https://www.nature.com/articles/s41586-022-05642-9). Eventually, in warm future climate, frequency will reduce if glaciers are very small to inexistant: if there's no glacier, it cannot produce a GLOF anymore… Although could indeed expect a rise at first with strongly changing glaciers and large amounts of melt: i.e., a bit like peak water concept for glaciers, but then instead for GLOFs.
- l.214-215: last sentence, in which you seem to directly make a link with climate change. But are we sure this is the case and that this event can (statistically) be attributed to climate change? It may be more likely due to climate change (from the limited evidence we have), but need to be careful to explicitly make this link. A bit in same line as collapse of Marmolada glacier last summer (e.g., EGU 2023 abstract by Gascoin and Berthier): could this event not have occurred without climate change? Difficult to make concluding statements about this without dedicated calculations and (many/detailed) field observations and measurements.

Thanks a lot for considering these comments. I look forward to receiving an updated (final?) version of your manuscript. And thank you once again for choosing 'The Cryosphere' for disseminate this interesting brief communication.

Best regards,
Harry Zekollari

---

## Author Response (AR2)

| TC-2022-237              **2nd revision** |
| :--- |
| **Brief Communication: Rapid ~335 $10^6$ m³ bed erosion after detachment of the Sedongpu Glacier (Tibet)**
Andreas Kääb, Luc Girod |

**2nd revisions in response to referees**

**General response**

We would like to thank again referee #2 for his careful and constructive comments and suggestions. We implemented almost all suggestions, with the exception of a very minor one regarding the naming of Brahmaputra vs. Yarlung Tsangpo.

Referee comments are in *italic*, and our response in normal font.

An annotated version of our revised manuscript with track changes is attached.

**Response to Referee #2, Max Van Wyk de Vries**

*I thank the authors for their rapid response to the two reviews and edits to the manuscript. The changes to the manuscript go a long way towards addressing my previous concerns, particularly by enhancing the discussion around the causes of the extremely rapid erosion episode and the broader implications of the work. I am glad to see this newly added sentence in the abstract "The recent erosion volumes at Sedongpu are by order of magnitude equivalent to the average annual denudation volume of the entire mountainous part of the Brahmaputra River basin, and illustrate a potential and intensity for rapid post-glacial landscape evolution and the hazards related to such high-magnitude low-frequency events that have rarely been considered so far." which I think will interest a whole new group of potential readers.*

Thanks

*Overall, I recommend the authors make minor revisions to complete a few of the additional changes described here, after which this manuscript would be suitable for publication in TC.*

Thanks

*If the intention was to create a full-length manuscript, I would have recommended that the authors add in a component of landscape modelling to the manuscript alongside the remote sensing. The two methods would complement each other nicely in evaluating this extreme event, and may allow for some constrains on the properties of the sediment (e.g. erodibility). This remote-sensing only manuscript makes for a good Brief Communication with the changes and does not have space for this added material, but it may be worth noting briefly in the discussions. It could make for a good follow-up paper to this.*

Added a sentence in the discussion: "Numerical modelling of the landscape evolution at Sedongpu could provide further constrains on the properties of the sediments and their mobilization but is beyond the focus of this brief communication."

*Finally, the new supplementary material is very useful and should be discussed in a little more detail. The Sentinel-1 timeseries shows the rapid unzipping of the landscape in a way that is not currently captured in the manuscript. In addition, the period of rapid change in the delta within the Tsangpo from June to Aug 2021 provides some clues into when sediment was being delivered to this river. I would like to see 1-2 more sentences describing this. I am not sure if there is space for a new figure in the manuscript, but showing the following three images really highlights the processes occurring during the period between the two DEMs (e.g. below). Maybe it could be a figure in the sup mat and referred to directly.*

We replaced Fig 3 (profiles) with Sentinel-1 images, added the profiles in a small version to Fig 2, added the full-size profiles to the Supplement and added text in the main paper: "The Sentinel-1 image time series over summer 2021 (Fig. 3 and animation in the Supplement) shows rapid changes of the Sedongpu fan in extent, shape and height, but still these changes appear rather minor compared to the 279±5 $10^6$ m$^3$ erosion volume that should have entered the fan during this time period."

and

"Another indication that supports this interpretation of gradual erosion is the fact that the fan of the Sedongpu valley in the Yarlung Tsangpo showed rapid changes during summer 2021 but seemed to have never dammed up the main river (Sentinel-1 images in Fig. 3 and the animation in the Supplement). Such damming happened after the 2018 glacier detachment."

[Figure]

*Finally, I am wondering about one other potential implication of this event. The volumes of sediment mobilized are on the same order as a very large landslide. This sediment happened to be delivered to the Tsangpo, one of the rivers with the greatest sediment transport capacity in the globe, so could largely be accommodated into the system. However, if this had occurred in a smaller catchment, there would be a very high chance of the river being temporarily dammed with associated outburst flood risk. It is somewhat speculative, but it could be useful to note this point in the broader implications.*

Added: "… hazards related to it from debris flows and impacts on rivers. For instance, only the very large sediment transport capacity of the Yarlung Tsangpo let the river accommodate the extreme short-term erosion volumes delivered to it without causing major river-damming."

*A few specific points:*

*L8 (and elsewhere) remove 'River', it is not needed.*

We left 'River' at the first occurrences of 'Yarlung Tsangpo' and 'Brahmaputra', but removed else.

*L15 'mountainous part of the…' Not quite clear what this means. Could you be more clear and reword?*

Modified to "…of the entire Brahmaputra basin upstream of the location where the river leaves the Himalayas, …"

*L42 maybe "Yarlung Tsangpo (also known as the Brahmaputra in its lower reaches)"?*

We prefer to keep as is. To our best knowledge and research, 'Brahmaputra' refers typically to the entire river, including the Tibetan reach. However, in Tibet the Tibetan reach (and only this one) is called 'Yarlung Tsangpo'. The Chinese literature refers typically to Yarlung Tsangpo. At the very end this naming question turns into a historical and political one, and even one about cultural appropriation. As written now, we try to be neutral to these questions.

*L52 'should have been' -> 'was'*

Changed

*L73-80 I understand that this material was added in response to the other review's question about removal of material in the initial event, but I am not very convinced by it. The ongoing destabilization of this glacier raises questions about many of the assumptions underpinning the ice-thickness calculations in both the Farinotti et al., 2019 and Millan et al., (2022) datasets. The problems may be more apparent in Millan et al.'s dataset, but the Farinotti et al., dataset may match the elevation loss by coincidence (examining the spread within the different models averaged at this location may give some idea). Finally, the uncertainties in both of these datasets for an individual glacier are much larger than the DoD and I am not sure about the usefulness of this comparison. You can mention it, but it will need to be framed by more discussion about the inherent uncertainties of these data.*

*Beyond this, I am not sure how much it matters whether the initial collapse was entirely composed of ice or entrained basal sediment for the remainder of this manuscript. If you say something along the lines of 'Pre-collapse ice-thickness datasets are not of sufficient accuracy to evaluate whether the initial event was entirely composed of glacier ice, whether it entrained basal sediment, and what the volume of sediment entrained might have been.*

*Examination of post-collapse optical imagery could not identify a large erosional scar in the subglacial sediment (Kaab et al., 2021), although this was not confirmed by direct field observations.'*

Added both suggested sentences in the manuscript.

*L103 I am genuinely astonished that this volume of material could be removed without the occurrence of debris flows. I only had time to have a very quick read through Yang et al.'s preprint, but do you have an*

*idea what scale of debris flow could have been missed by their equipment? It sounds like it was moved to a point higher in the channel in 2021 following the March event, so may have been less sensitive?*

We investigated closer and modified the text: "In fact, state-of-the-art early warning installations including cameras and geophones at the outlet of the Sedongpu valley registered rock-ice avalanches (following section) but no massive debris flows from the former glacier bed and no river blockings of the Yarlung Tsangpo are reported (Yang et al., 2023).However, a new early warning station further up in the Sedongpu valley was only installed in May 2022, and then also able to detect debris flows from the catchment."

*L143-150 It would be good to refer to the Sentinel-1 imagery in this, as it supports the description (which appears a little speculative without it).*

Done

*L180-183 Again, really happy to see this larger-scale description here, which is one of the most remarkable findings in my view. This sentence needs a reference (or several) for the source of the basin-wide erosion rate data.*

Done. Added also a sentence on long-term uplift/denudation rates in the region in the study site description.

*L192-193 The two halves of this sentence are not entirely equivalent. While the volume and rate of the Sedongpu erosion dwarf GLOF, the relative frequency of each is (as far as I know) not known. This should be added to the sentence or reworded so it is not implying that these events are an even larger driver of erosion in the Himalaya (which I don't think is what you are trying to say).*

Reworded to "Lake outburst floods have been suggested to be major drivers of erosion in the Himalayas (Cook et al., 2018). The erosion volumes and rates at Sedongpu dwarf even those from lake outburst floods, though it is unclear how the frequency of both event types and thus their long-term volumes relate to each other.

*Code availability: This change is good and it makes it easier for readers to find the exact information.*

---

## Author Response (AR3)

| TC-2022-237 | **3ʳᵈ revision** |
| --- | --- |

**Brief Communication: Rapid ~335 10⁶ m³ bed erosion after detachment of the Sedongpu Glacier (Tibet)**
Andreas Kääb, Luc Girod

**3ʳᵈ revisions, response to editor**

Editor comments in *blue italic*, author responses in normal font. (Combined font characters did unfortunately not translate correctly from the pdf of the editor comments into the below text).

*Thanks a lot for having responded to the new comments during the second round of revisions and for having updated the manuscript accordingly. When comparing the current manuscript to the version at the stage of initial submission (December 2022), I think it is fair to say that the manuscript has improved in clarity and that the additional details that you added make the story very interesting, also for those who are not directly in the field of glacier geomorphology/erosion/hazards (like me). I am convinced that this short, yet very clear story will be of interest to the readers of The Cryosphere. After reading the latest version of your manuscript, I have formulated a series of mostly minor and easy to incorporate suggestions that I hope you will find helpful. I invite you to consider these comments, after which we should normally be able to proceed to the final acceptance of your manuscript.*

We would like to thank the editor for the careful and detailed comments and suggestions on this version of the manuscript. Below are our point-by-point responses.

- *l.12: probably best to have a consistent use of m³ or km³ to make quantites directly comparable. Not only here, but throughout the manuscript.*
We put 0.6 km³ in brackets after 600 10⁶ m³ for non-expert readers less familiar with millions of m³. This should have been the only km³ unit in the manuscript.

- *l.15-16: this last sentence of the abstract was quite difficult to understand. Consider rewriBng to something along the lines of: "…the Himalayas. This high-magnitude low- frequency event illustrates a potenBal for rapid post-glacial landscape evoluBon and associated hazards that have rarely been observed (at such high intensity) so far".*
Implemented

- *l.19: "…disappearance, the̲se n̲ewly uncovered areas are…"*
Done

- *l.23: "…comparably slow̲ly, over…"*
Done

- *l.26: not sure you need "respecBvely" here, since you do not refer to anything*

*menBoned before in a given order. Would suggest removing this here. Same for occurrence on l. 67.*

Done

- *l.29: "…indicaBon on the maximum…"*

Done

- *l.30: "…detachment, enBre…" (also other occurrences where a "," would be needed: e.g., in l. 53 "Obu et al. (2019), …"), l.99 ("..study site, only very…")*

Done, and we rely on the proof reading to find other occasions.

- *l.36-38: hard to understand. Possibly change to: "We summarize key site informaBon on the 2018 glacier detachment, and quanBfy the glacier-bed volume changes and other landscape changes in the basin unBl 2022" (possibly even add unBl which month in 2022)*

Done

- *l.40: for the study site descripBon, in the current formulaBon, it seems like there is no glacier remaining at all? While in reality a part of the glacier survived / did not collapse? Would be good to specify this a bit more. Also, to frame it becer, maybe start the sentence with: "At the Bme of its detachment, the Sedongpu glacier was…"*

Clarified in section 2 and 4 that the tongue/lower glacier part detached. Reformulated as suggested.

- *l.40: elevaBon of about 3700 m: could you provide the elevaBon range of the glacier at the Bme of detachment? And possibly also for what is now remaining of the glacier?*

Included in sections 2 and 4, respectively.

- *l.42: "… The highest point"*

Done

- *l.43-44: extreme angles of the slopes: could you provide a quanBficaBon for this statement? What slope for the angles are we to expect here?*

Done (40-45 deg)

- *l.45: "…Tsangpo has an…"*

Done

- *l.52: "The wider study region…"*

Done

- *l.68: "…avalanches ran from the Gyala west flank over…"*

Done

- *l.72: "enBre tongue": so from this I tend to understand that the enBre glacier did not collapse? See related comment above. Would be good to have a quanBtaBve indicaBon about how much of the glacier was lost and e.g., the elevaBon range of the glacier before and aKer the collapse.*

Done

- *l.72: possibly reword to: "…detached, complemented by an addiBonal…"*

Done

- *l.82: yes, indeed quite high uncertainBes for the ice thickness reconstrucBon. Aside from the change in velocity, the fact that relaBve errors are very large for velociBes of slow-flowing glaciers also leads to a large (relaBve) error in the corresponding ice thickness reconstrucBon by Millan et al. (2022). Would be good to menBon this in one or two addiBonal sentences.*

Done

- *l.89: "unBl 2022" (add white space)*

Done

- *l.88-92: quite a long and fragmented sentence. Suggest splifng this up in two sentences, e.g., "and its ,surroundings, with maximum erosion depth of 360 m and an average of 135 m over an area of 2.5 km², amounBng to about 335+-5 10⁶ m³. This volume corresponds to about 2.5 Bmes the detached glacier volume (Figs. 1-2…"*

Done

- *l.92: "…can be observed at limited…"*

Done

- *l.93: "…elevaBon changes from January…"*

Done

- *l.94: "…contribute by far to the largest…"*

Reformulated

- *l.110-111: "… (Yang et al., 2023). A new early…May 2022, and was then also…"*

Reformulated

- *l.118: glacier bed being "likely temperate": is there any evidence for this statement? Measurements and/or modelling of glaciers in this region? Would be good to specify and provide addiBonal info for this.*

We removed this statement. It was based on our interpretation of the climatology of the region, the regional permafrost limit, and the lack of continuous ice flow from the highest (= potentially coldest) parts of the catchment. (Nourishing of the glacier tongue is rather through avalanching). We prefer to remove the statement instead of adding a long discussion on the topic.

- *l.127: "contributed to the ice and sediment properBes in the valley": sounds a bit vague/mysterious here: can this be reformulated to be more specific? Or possibly remove this? (the sentence also works well without this)*

Reformulated

- *l.132: maybe reword to "…was able to transport most of the…": i.e. omit "further"*

Done

- *l.136: "It would be interesBng…"?*

Done

- *l.141: "In summary, between early 2017 and November 2022, around…": and ideally, be even more specific for what early 2017 is (i.e., which month)*

Done

- *l.141: 659 +- 7: this +-7 remains a remarkably small error esBmate (i.e., a mere 1% of the total volume)…*

Not really, 7 Mm3 is quite a volume uncertainty and in itself equivalent to a huge event. It just appears small relative to the giant volume mobilized from Sedongpu. From the high-res satellite stereo data used such accuracy is well expected and not a surprise. The elevation accuracy does not scale with the lost volume, it remains largely constant.

- *l.141-143: suggest splifng up in two separate sentences: "…bedrock and sediments. About half of Bme volume (335+-5 $10^6$ $m^3$) is esBmated to be eroded from the…"*

Done

- *l.144: "…in the lacer volume": what is this exactly? Can you be more specific here?*

Clarified

- *l.151: "…could be parBcularly prone to erosion. This…" + on l.162: "…bed was much more prone to erosion than the…"*

Done

- *l.160: "…sediments, which are perhaps…"*

Done

- *l.169: very licle precipitaBon. Has this been quanBfied, and could you provide a figure for these numbers? e.g., how this compared to other (standard) years, with this year having for instance X% less precipitaBon?*

We reformulated. This refers to the Suppl. Figures. The main point is that none of the datasets consulted gives an indication of strong precipitation that could have driven the erosion. Relative amounts compared to other years are less important in this context. And, given the uncertainties of all these data sets for the remote and extreme mountain topography of the study region, we prefer to not quantify precipitation amounts from these.

- *l.172: "…terrain gradients. Numerical modelling": i.e., suggest removing the "though" here.*

Reformulated

- *l.174: other glacier detachments. Can you menBon here how many detachments these are? e.g., "…detachments (X in total) listed in Kääb et al. (2021)"*

Done

- *l.175: suggest rewording to: "…, we do not find as important extreme erosion in these other cases compared to Sedongpu, but…"*

Done

- *l.178: "…potenBally pronounced soK sediments…"*

Done

- *l.180: "…most glacierized mountains on Earth": reference for this statement?*

Done

- *l.200: when calculaBng the size of the hypotheBcal catchment, you may want to refer to how*

*much larger this is than the actual catchment, e.g., "...catchment (X Bmes more than actual size of the catchment)"*

Clarified that the 250,000 km2 is actually the actual mountain catchment size of Brahmaputra, not a hypothetical size.

We also added one more sentence with another equivalent for the Sedongpu erosion volume, in late response to ref #2 who asked for long-term/large-scale perspectives of the Sedongpu volumes: *Multiplying the Sedongpu Glacier catchment area (50 km$^2$) by vertical motion rates of 5 mm yr$^{-1}$ (Zhao et al., 2023) gives an uplifted volume of 250 10$^6$ m$^3$ 1000 yr$^{-1}$ indicating that the rock and sediment volumes recently eroded from Sedongpu are roughly equivalent to the volumes uplifted over 1–2 millennia for the entire catchment, neglecting density differences.*

- *l.211: unclear what signal is in change of GLOFs. Would be worth menBoning recent study in Nature by Veh et al. (2023), who suggest that GLOFs are reducing in frequency ([hcps://www.nature.com/arBcles/s41586-022-05642-9](hcps://www.nature.com/arBcles/s41586-022-05642-9)). Eventually, in warm future climate, frequency will reduce if glaciers are very small to inexistant: if there's no glacier, it cannot produce a GLOF anymore... Although could indeed expect a rise at first with strongly changing glaciers and large amounts of melt: i.e., a bit like peak water concept for glaciers, but then instead for GLOFs.*

We prefer to leave the (vague, admitted) text as is in order to not discuss glacier lake outburst flood frequency over time (which time-scale?), as this is a complex issue that we don't want to rise at the very end of our brief communication. Veh et al (2023) refer to ice-dammed lakes. For moraine-dammed lakes, Harrison et al. (2017; https://doi.org/10.5194/tc-12-1195-2018) suggest a lagged increase in response to glacier shrinkage. This topic is currently debated in the community (e.g. also Veh et al. 2019; doi: 10.1038/s41558-019-0437-5). In response to a referee, we want here to just mention that the observed erosion volumes can also be seen in comparison to lake outbursts.

- *l.214-215: last sentence, in which you seem to directly make a link with climate change. But are we sure this is the case and that this event can (staBsBcally) be acributed to climate change? It may be more likely due to climate change (from the limited evidence we have), but need to be careful to explicitly make this link. A bit in same line as collapse of Marmolada glacier last summer (e.g., EGU 2023 abstract by Gascoin and Berthier): could this event not have occurred without climate change? Difficult to make concluding statements about this without dedicated calculaBons and (many/detailed) field observaBons and measurements.*

Clarified that we refer to the disappearance of a glacier rather than directly to climate change. We believe our study shows an extreme case of what erosion volumes and speeds can develop after glacier loss.

*Thanks a lot for considering these comments. I look forward to receiving an updated (final?) version of your manuscript. And thank you once again for choosing 'The Cryosphere' for disseminate this interesBng brief communicaBon.*

Thank YOU!

**From the Copernicus team:**

*Notification to the authors:*

*Regarding the satellite, drone, or airborne images in Figure 3: If you are not the originator of the images, then appropriate credit or copyright must be given (https://publications.copernicus.org/for_authors/manuscript_preparation.html#mapsaerials). If applicable, please add the necessary details to the figure or the figure caption for the next revision.*

Done